# Highly-conducting molecular circuits based on antiaromaticity

Shintaro Fujii[1], Santiago Marqués-González[1], Ji-Young Shin[2], Hiroshi Shinokubo[2], Takuya Masuda[3], Tomoaki Nishino[1], Narendra P. Arasu[4], Héctor Vázquez[4] & Manabu Kiguchi[1]

Aromaticity is a fundamental concept in chemistry. It is described by Hückel's rule that states that a cyclic planar $\pi$-system is aromatic when it shares $4n + 2$ $\pi$-electrons and antiaromatic when it possesses $4n$ $\pi$-electrons. Antiaromatic compounds are predicted to exhibit remarkable charge transport properties and high redox activities. However, it has so far only been possible to measure compounds with reduced aromaticity but not antiaromatic species due to their energetic instability. Here, we address these issues by investigating the single-molecule charge transport properties of a genuinely antiaromatic compound, showing that antiaromaticity results in an order of magnitude increase in conductance compared with the aromatic counterpart. Single-molecule current–voltage measurements and *ab initio* transport calculations reveal that this results from a reduced energy gap and a frontier molecular resonance closer to the Fermi level in the antiaromatic species. The conductance of the antiaromatic complex is further modulated electrochemically, demonstrating its potential as a high-conductance transistor.

[1] Department of Chemistry, Graduate School of Science and Engineering, Tokyo Institute of Technology, Ookayama, Meguro-ku, Tokyo 152-8551, Japan. [2] Department of Molecular and Macromolecular Chemistry, Graduate School of Engineering, Nagoya University, Aichi 464-8603, Japan. [3] Global Research Center for Environment and Energy Based on Nanomaterials Science (GREEN), National Institute for Materials Science (NIMS), Tsukuba 305-0044, Japan. [4] Institute of Physics, Academy of Sciences of the Czech Republic, Cukrovarnicka 10, Prague CZ-162 00, Czech Republic. Correspondence and requests for materials should be addressed to S.F. (email: fujii.s.af@m.titech.ac.jp) or to H.S. (email: hshino@chembio.nagoya-u.ac.jp) or to H.V. (email: vazquez@fzu.cz) or to M.K. (email: kiguti@chem.titech.ac.jp).

The concept of aromaticity[1–3] is of fundamental importance in the chemistry of π-electrons that play an essential role in many organic materials relevant for nanoscale chemical, physical and biological studies. The stability and electronic properties of the π-system depend sensitively on the geometry and connectivity of the molecule. For instance, when a linear π-conjugated system adopts a cyclic and planar framework, the molecule gains additional energetic or resonance stabilization. This resonance or aromatic stabilization energy represents the energy difference with the best canonical structure that can be drawn for the molecule and is thus a measure of molecular stability. The greater this energy, the more stable the compound is. In addition to topology, this quantity also depends on the number of π-electrons in the cyclic conjugated system. Cyclic and planar molecules with $4n + 2$ π-electrons present the highest resonance energy (stability) and are referred to as aromatic within Hückel's rule[4]. On the other hand, cyclic and planar molecules sharing $4n$ π-electrons feature a remarkable energetic destabilization and are referred to as antiaromatic[1,5]. For example, the lower resonance energy in cyclobutadiene makes it more energetic (and far more reactive) than its open-chain counterpart 1,3-butadiene. Higher stability and lower reactivity are reflected in larger molecular gaps and aromatic molecules generally have larger gaps than antiaromatic ones[6,7]. From a fundamental and simplified viewpoint, cyclic molecules with $4n + 1$ and $4n + 3$ π-electrons obviously possess unpaired electron(s) and become nonaromatic. These compounds exhibit open shell character and cannot be chemically isolated in a controlled manner.

Aromatic and antiaromatic π-systems have attracted wide attention in the field of the organic synthesis and materials science[8–11]. Aromatic molecules such as oligoacenes[8], porphyrins[9], tetrathiafulvalenes[10,11] and related compounds play an important role in organic electronic components such as electroluminescent devices and organic photovoltaic devices. In contrast, the characteristic instability and smaller gap of antiaromatic molecules brings to the π-system a great ability to donate and accept charge carriers and eases electrochemical tunability of the molecular orbitals, making antiaromatic compounds well suited for organic electronic components. Indeed, antiaromatic compounds had been predicted by Breslow to be more conducting. Early electrochemical measurements of oxidation/reduction potentials had shown that electron transfer from donor to acceptor groups in a molecule was amplified when connected through an antiaromatic structure[12,13]. Advances in electronic characterization make it possible to measure an electronic current through individual molecules, probing their inherent charge transport properties at the single-molecule scale[14–16]. Recent single-molecule electronics studies of a series of molecules with varying degrees of aromaticity established a negative relation between aromaticity and conductance. More aromatic rings with higher resonance energies were seen to correlate with HOMO (highest occupied molecular orbital) positions further from the Fermi level and lower conductance (which in these molecules is HOMO dominated)[17]. Subsequent transport calculations verified this trend and reported smaller molecular gaps and HOMO positions closer to the Fermi level with decreasing aromaticity[18]. However, the chemical instability of the antiaromatic molecules and unfavourable intermolecular interaction in the bulk has restricted transport studies to aromatic systems or species with no aromaticity. While these works clearly established the negative relationship of aromaticity with conductance, the positive relationship of conductance and antiaromaticity remained to be demonstrated[19].

The present study aims to address the long-standing issue of the electronic properties of antiaromatic molecules by investigating the single-molecule transport properties of an antiaromatic compound and exploring the effect of antiaromaticity on the charge transport properties at the single-molecule scale. To that end, a recently developed, antiaromatic norcorrole-based Ni(II) complex is studied (Fig. 1). Ni(nor) features a chemically stable, structurally rigid antiaromatic porphyrinoid moiety[20–22] where antiaromaticity can be preserved due in no small part to its molecular rigidity. The electronic properties of Ni(nor) are benchmarked against a porphyrin-based aromatic counterpart, Ni(porph) (Fig. 1). The antiaromatic molecule Ni(nor) exhibits enhanced charge transport properties in which single-molecule conductance is more than one order of magnitude higher than that of its aromatic counterpart. This makes Ni(nor) the most conductive organometallic complex reported to date, to the best of our knowledge. The *ab initio* transport calculations show that the remarkable conductance of Ni(nor) originates from the smaller gap and more favourable frontier orbital alignment with respect to the aromatic species. Through electrochemical gating, the conductance of the antiaromatic molecule Ni(nor) is further modulated by more than a factor of five. This work ultimately expands the current synthetic guidelines for the design of highly conducting and functional organic electronic materials using antiaromaticity.

## Results

**Conductance studies.** To assess the impact of antiaromaticity in molecular conductance, single-molecule transport studies were performed on two structurally similar complexes, Ni(nor) and Ni(porph) (Fig. 1, Supplementary Figs 1,2, and Supplementary Note 1). The thioacetate groups bind to Au electrodes through the S lone pair, a strategy that has been shown to result in selective bonding and clear experimental conductance signatures (see Supplementary Note 3 for further details)[23]. The scanning tunnelling microscope-based break-junction (STM-BJ) technique was used to obtain 2,500 conductance–distance traces for each molecule (see Methods and Supplementary Fig. 3). Measurements were repeated on analogue samples of both complexes to ensure reproducibility. All recorded traces were employed to build logarithmically binned conductance histograms with no data selection (Fig. 2a). Conductance histograms display well-defined peaks denoting the most probable molecular conductance value at $4.2 \times 10^{-4}$ and $1.7 \times 10^{-5}\,G_0$ for Ni(nor) and Ni(porph), respectively. No additional molecular signatures were observed in the high conductance regime $10^{-2}$–$1\,G_0$ (Supplementary Fig. 5). The measured conductance of the antiaromatic complex Ni(nor) was found to be an order of magnitude higher than that of Ni(porph). The comparatively low conductance of the aromatic counterpart is in good agreement with previous studies on closely related porphyrin-based complexes[24,25]. The structural similarities between both complexes suggest that the enhanced conductance of Ni(nor) arises from the antiaromatic nature of the norcorrole moiety. A two dimensional conductance–distance histogram constructed with traces featuring Ni(nor) molecular plateaus extracted from the same data set reveals junction lengths of ∼0.8–1.0 nm (Fig. 2b; Supplementary Figs 6 and 7), ruling out the possibility of molecular junctions being formed perpendicularly to the norcorrole plane[26]. Slightly longer junction dimensions of ∼1.0–1.2 nm were typically observed for Ni(porph) (Supplementary Fig. 8) in good agreement with previous reports[25]. Similar junction lengths were obtained for both complexes at different voltages (Supplementary Figs 9 and 10). The comparatively shorter length of Ni(nor) junctions reflects the marked structural constraints of the antiaromatic norcorrole moiety, in sharp contrast with porphyrin moiety known to adopt both buckled and twisted conformations (Supplementary Fig. 11)[20,27,28]. Additional STM-BJ experiments were performed in ultrahigh vacuum (UHV) to confirm the

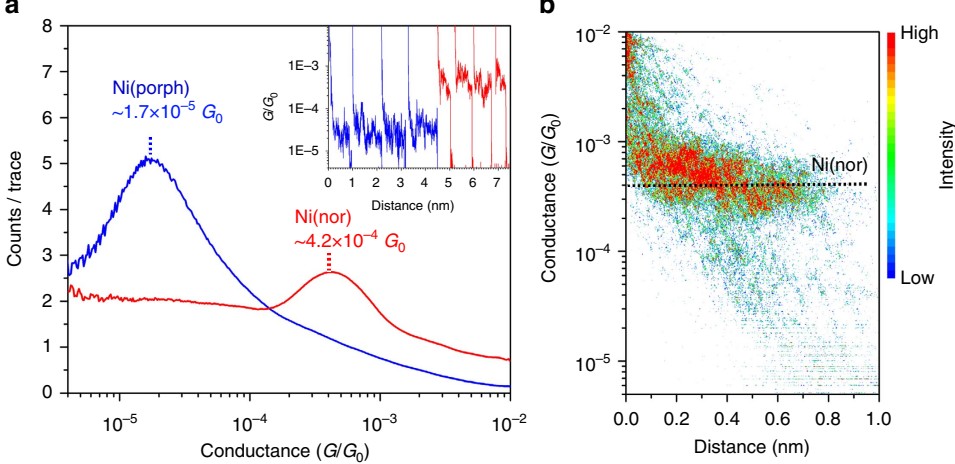

**Figure 1 | Antiaromatic and aromatic molecules.** Antiaromatic Ni norcorrole Ni(nor) and its porphyrin-based aromatic counterpart Ni(porph) (Supplementary Note 1).

**Figure 2 | Measured single-molecule conducting properties.** (**a**) Logarithmically binned conductance histograms (111 bins per decade) of Ni(nor) and Ni(porph) at 100 mV in air. The peaks at $4.2 \times 10^{-4}$ and $1.7 \times 10^{-5} G_0$ denote the most probable molecular conductance values of Ni(nor) and Ni(porph), respectively. Histograms were built from 2,500 conductance traces with no data selection. Inset: representative examples of STM-BJ conductance–distance traces of Ni(nor) and Ni(porph) featuring molecular conductance plateaus. Traces were laterally offset for clarity. (**b**) Two-dimensional (2D) conductance–distance histogram of Ni(nor). Traces showing a conventional tunnelling decay were removed for clarity. For additional conductance measurements and histograms see Supplementary Figs 4–11.

remarkable charge transport properties and chemical stability observed for Ni(nor). In UHV, a well-defined molecular conductance peak was observed at $4.6 \times 10^{-4} G_0$ (Supplementary Note 9), demonstrating the robustness of the electronic transport probabilities of the chemically stable antiaromatic compound of Ni(nor).

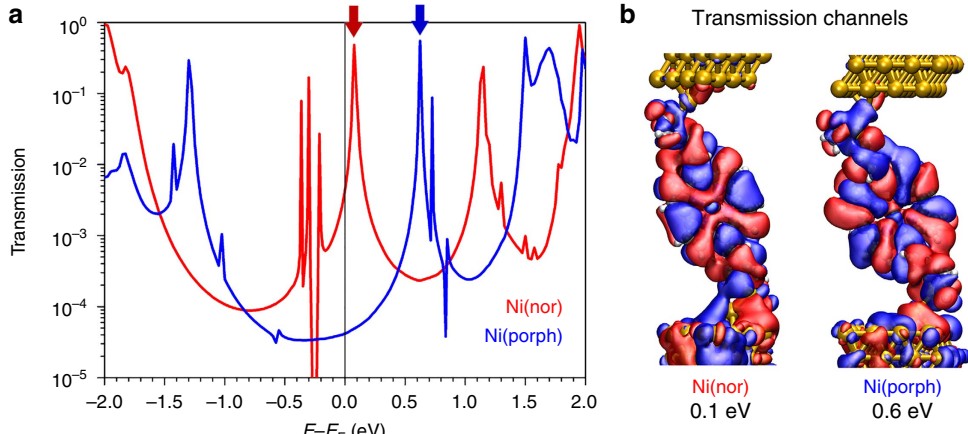

**Figure 3 | Calculated transport properties.** (**a**) DFT transmission spectra of Ni(nor) and Ni(porph). Peaks at 0.1 and 0.6 eV are indicated by arrows. (**b**) Real part of the most conducting channels of Ni(nor) at 0.1 eV and Ni(porph) at 0.6 eV. In both cases, current is carried by a single channel.

**Table 1 | Comparison of resonance-corrected calculated and measured conductance (in units of $G_O$) and molecular-level offsets (in eV) of Ni(nor) and Ni(porph).**

|  | Ni(nor) | | Ni(porph) | | Ratio G(nor)/G(porph) |
|---|---|---|---|---|---|
|  | $E_{LUMO}$-$E_F$ | $G(E_F)$ | $E_{LUMO}$-$E_F$ | $G(E_F)$ |  |
| Calculations | 0.47 | $2.1 \times 10^{-4}$ | 0.93 | $1.5 \times 10^{-5}$ | 14 |
| Experiment | 0.48 | $4.2 \times 10^{-4}$ | 0.85 | $1.7 \times 10^{-5}$ | 25 |

**Transport calculations**. To clarify the origin of the higher conductance of the antiaromatic Ni(nor) complex and understand the effect of antiaromaticity on the charge transport properties, transmission calculations based on density functional theory (DFT) and nonequilibrium Green's functions, including corrections to the DFT resonance positions, were performed (see Supplementary Notes 4,5 and Supplementary Figs 12–15). As discussed in the previous section, the Ni(nor) and Ni(porph) molecules bind to Au electrodes using acetyl groups at the molecular termini and form single-molecule junctions (see also Supplementary Note 4). Figure 3a shows the DFT spectra of the Ni(nor) and Ni(porph) species. The spectrum of the Ni(porph) complex shows a peak slightly above $\sim +0.6$ eV while that of the Ni(nor) features a low-lying peak at $\sim +0.1$ eV. For both Ni(porph) and Ni(nor), zero-bias conductance is determined by an unoccupied molecular frontier orbital that tails into the Fermi level and the current is carried by a single transmission channel. The width of the transmission peaks measures the degree of electronic coupling of the conducting orbital to the electrodes. From the spectra, we obtain full-width at half-maximum of $\Gamma_{Ni(nor)} = 16$ meV and $\Gamma_{Ni(porph)} = 10$ meV (Supplementary Table 2). Analysis of the most conducting transmission channel at that energy shows that it is a state delocalized over the conjugated core and linker rings (Fig. 3b) deriving from the lowest unoccupied molecular orbital (LUMO). It should be noted that the series of peak-dip features in the occupied part of the Ni(nor) spectrum close to the Fermi level are very sharp and do not influence zero-bias conductance (Supplementary Note 4).

It is well known that DFT calculations inherently overestimate conductance due to errors in the position of frontier molecular orbitals[29–33]. Corrections to the positions of DFT molecular resonances, which include a molecular self-interaction term as well as a contribution arising from the polarization of the interface,

were applied to the Ni(nor) and Ni(porph) junctions (details are available in Supplementary Note 5 and Supplementary Table 1). Table 1 summarizes these results and compares the measured conductance to the calculated values after the corrections to the DFT molecular-level positions are taken into account. The corrections to the molecular resonance positions bring the calculated values into very good agreement with experiment, as previously seen for porphyrins[25]. Calculations show that the higher conductance of the antiaromatic species results from a LUMO-derived resonance closer to the Fermi level.

**Current–voltage ($I$–$V$) measurements**. To confirm the theoretical prediction that the higher conductance of the antiaromatic compound Ni(nor) is due to the energetically closer lying resonance, the $I$–$V$ characteristics of Ni(nor) and Ni(porph) were measured using a variation of the STM-BJ technique. Briefly, upon formation of a single-molecule junction, the tip is momentarily held in position and the tunnelling current is monitored as the voltage is swept in the $\pm 1$ V range (see Methods, Supplementary Fig. 16 and Supplementary Notes 6–8 for details). The process is repeated until a statistically significant data set is obtained. The selected profiles were compiled into logarithmically binned two-dimensional plots to obtain a statistical description of the $I$–$V$ characteristics of Ni(nor) and Ni(porph) (Fig. 4a,b). The steeper slope of the Ni(nor) $I$–$V$ profiles in the linear regime $\pm 0.1$ V ($I/V = \sim 4 \times 10^{-4} G_0$) is in good agreement with the aforementioned STM-BJ studies. The nonlinear $I$–$V$ response is evident in the $I$–$V$ profiles at the high bias voltage-range above 0.5 V. Each nonlinear $I$–$V$ profile was individually fitted to a single-level tunnelling model[34,35] described by

$$I(V) = \frac{2e\Gamma}{h} \left\{ \tan^{-1}\left[\frac{eV/2 - \varepsilon_0}{\Gamma}\right] + \tan^{-1}\left[\frac{eV/2 + \varepsilon_0}{\Gamma}\right] \right\}, \quad (1)$$

where $\varepsilon_0$ is the energy difference between the chemical potential of

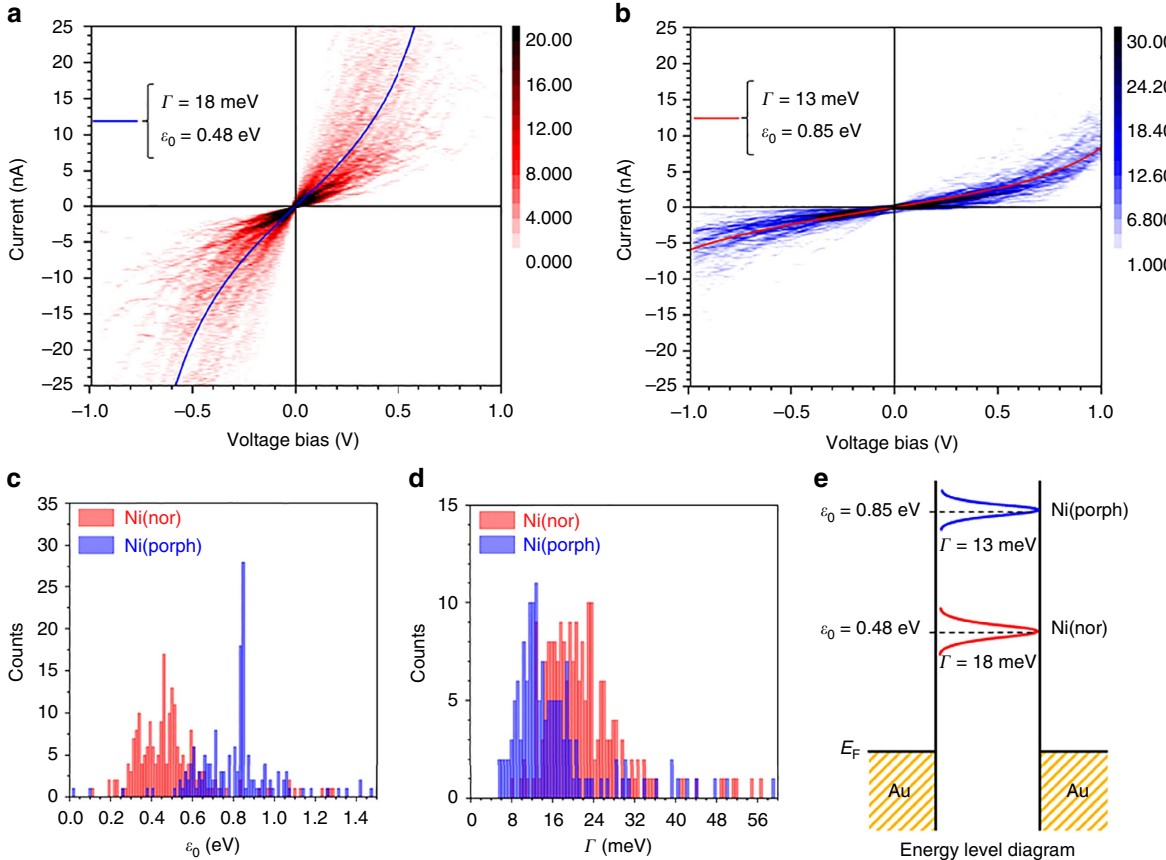

**Figure 4 | Experimental determination of transmission resonance positions and electronic couplings.** (**a**) Two-dimensional (2D) histogram built from 216 I–V profiles collected from single-molecule junctions of Ni(nor). (**b**) Current–voltage histogram constructed with 168 profiles of Ni(porph). Overlaid lines show the most probable I–V profile. All current–voltage profiles were recorded during single-molecule conductance plateaus in the correlated BJ trace. Bin sizes of 0.02 V and 0.075 nA were employed. (**c**) Statistical distribution of transmission peak position Ni(nor) and Ni(porph) extracted from the fitting of each individual I–V response to a single-level tunnelling model. (**d**) Statistical distribution of metal–molecule coupling for Ni(nor) and Ni(porph) single-molecule junctions. (**e**) Schematic representation of the orbital alignment and metal–molecule electronic coupling for Ni(nor) and Ni(porph) assuming a single-level model.

the electrode and the dominant molecular transport orbital and $\Gamma$ is the metal–molecule coupling (full-width at half-maximum) (see Supplementary Figs 17–19). The statistical distribution of $\varepsilon_0$ and $\Gamma$ for Ni(nor) and Ni(porph) is shown in Fig. 4b,c. Both complexes show a comparable degree of metal–molecule electronic coupling with $\Gamma \sim 18$ meV and $\sim 13$ meV for Ni(nor) and Ni(porph), respectively. The experimentally obtained electronic couplings are in very good agreement with the theoretical ones of 16 and 10 meV for Ni(nor) and Ni(porph), respectively. The similarity between values obtained for the metal–molecule couplings of both complexes can be rationalized in terms of their structural similarity and the use of identical thioacetate linkers. These results confirm the weakly binding nature of the thioacetate linkers, compared with previously reported studies on 1,4-benzenedithiol featuring the archetypal-SH linkers (10–130 meV)[36]. The moderate metal–molecule interaction displayed by the thioacetate moiety plays a crucial role in the preservation of the antiaromatic character of the complex, avoiding possible counterbalancing electronic effects[19,37] The energy difference between the Fermi level and the dominant molecular transport orbital was found to be much smaller ($\varepsilon_0 \sim 0.48$ eV) for the antiaromatic compound Ni(nor) than for the aromatic compound Ni(porph) ($\varepsilon_0 \sim 0.85$ eV). The I–V analysis indicates that the origin of the increased conductance of Ni(nor) is the closer-lying frontier orbital ($\varepsilon_0 \sim 0.48$ eV) that is directly related to the narrow HOMO–LUMO gap caused by the antiaromatic destabilization of the norcorrole moiety.

**Electrochemical conductance modulation.** The ambient and UHV STM-BJ studies and transport calculations described above revealed the LUMO-mediated enhanced charge transport capabilities of Ni(nor). Here we demonstrate tunability of the single-molecule conductance of Ni(nor) by electrochemically gating the position of this orbital with respect to the electrode potential.

Before single-molecule studies, cyclic voltammetry of Ni(nor) adsorbed on Au(111) was measured. The first electrochemical reduction, attributable to a reaction from Ni(nor) to Ni(nor)$^{\bullet-}$ (ref. 21), appeared at $-0.65$ V versus Ag/AgCl (Supplementary Note 10). Subsequent electrochemical single-molecule conductance experiments brought the LUMO-mediated transport channel closer to the electrode potential as the electrochemical potential was changed from $+0.45$ to $-0.30$ V versus Ag/AgCl. The STM-BJ technique was employed to collect conductance–distance traces of Ni(nor) in a pH 7 aqueous phosphate buffer at different electrochemical potentials ($E$) while keeping the tip–substrate voltage constant at 0.1 V (see Methods and Supplementary Figs 20 and 21 for details). The accessible electrochemical potential window was determined experimentally ensuring that the background tunnelling current would not affect the single-molecule measurements.

Logarithmically binned histograms show an increase in the molecular conductance of Ni(nor) as $E$ takes more negative values in the reduction direction and display well-defined peaks shifting from $6.9 \times 10^{-4} G_0$ at $E = -0.30$ V to $1.3 \times 10^{-4} G_0$ at $E = +0.45$ V (Fig. 5a and Supplementary Fig. 21). Under the

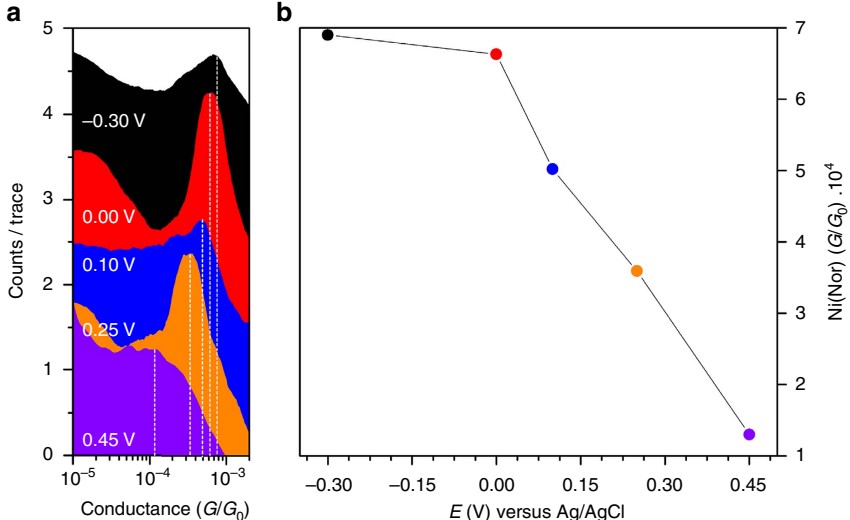

**Figure 5 | Electrochemical gating of the charge transport properties.** (**a**) Logarithmically binned conductance histograms of Ni(nor) built from STM-BJ trances at different electrochemical potentials. An approximately fivefold conductance modulation was attained for the antiaromatic complex Ni(nor) under electrochemical environment. Histograms were built from STM-BJ traces collected at a different electrochemical potential ($E$), while holding a constant tip–substrate bias of 0.1 V. Experiments were performed using a Ag/AgCl reference ($\Delta E_{ref} = \pm 1 mV$). (**b**) Ni(nor) conductance as a function of electrochemical potential. For details on the experimental setup, individual histograms and reproducibility see Supplementary Figs 20–24.

electrochemical potential control, the single-molecule conductance of Ni(nor) exhibits a fivefold modulation. Comparable results were obtained on an analogue sample (Supplementary Figs 23 and 24). In both cases, a substantial conductance modulation was observed for Ni(nor) (Fig. 5b and Supplementary Fig. 23). The measured conductance modulation results from the movement of the LUMO position of the neutral species with respect to the Fermi level of the electrodes.

The observed behaviour with the quasi-sigmoidal shape of the conductance modulation is similar to one of two distinct profiles recurrently found in the literature for both organic and organometallic derivatives in aqueous electrolytes[38–40]. A sigmoidal shape of the molecular conductance modulation has been observed previously in studies of electrochemically gated conductance of viologen-based single-molecule junctions in aqueous electrolytes and attributed to a 'soft' electrochemical gating of the conductance response[40,41]. A second behaviour, less frequently reported, features a bell-shaped conductance modulation profile[40,42] with a conductance peak centred at a threshold electrochemical potential that predicted to be shifted by typically a few hundred mV (corresponding to the reorganization energy of the molecule) from the redox potential[43]. Although both modulation profiles (that is, sigmoidal and bell-shaped profiles) have been rationalized for both direct tunnelling and redox-mediated hopping transport mechanisms, the origin of the sigmoidal profile remains a matter of debate. The absence of a maximum in the modulation profile within the potential range (from $+0.45$ to $-0.3$ V versus Ag/AgCl) may suggest that the resonance between the Fermi level and the Ni(nor) orbitals has not been reached. Recent studies have demonstrated that the sigmoidal profile may arise from the broadening of the bell-shaped gating response[40]. This result was rationalized in terms of the comparatively low gating efficiency of the aqueous electrolytes, roughly 20% that of the ionic liquid electrolytes. By replacing the aqueous electrolyte for an ionic liquid electrolyte the gating profile of a viologen derivative changed from sigmoidal to bell shaped[40]. While the origin of the profile in the electrochemical modulation remains to be solved, the present study revealed that the antiaromaticity is preserved in

the aqueous electrochemical conditions and Ni(nor) displays well-defined single-molecule conductance that enabled us to attain a fivefold conductance modulation.

## Discussion

We have used STM-BJ and $I$–$V$ measurements, supported by theoretical calculations, to explore the effect of antiaromaticity on the charge transport properties of single-molecule junctions. The rational molecular design of the antiaromatic Ni(nor) and aromatic Ni(porph) featuring weakly interacting thioacetate linkers enabled direct comparison between the single-molecule junctions of antiaromatic and aromatic characters. We compared an antiaromatic compound with its aromatic counterpart and showed that the antiaromatic species is an order of magnitude more conducting. Antiaromatic destabilization results in a smaller molecular gap and drives the LUMO closer to the Fermi level. The conductance of the antiaromatic species was further modulated by a factor of five by means of electrochemical gating. Our work demonstrates the potential of analogue derivatives as highly conducting molecular materials and provides relevant guidelines for the design of molecular materials for highly conducting single-molecule electronics.

## Methods

**Synthesis and characterization.** The general procedure to synthesize Ni(nor) and Ni(porph) is as follows (see the Supplementary Note 1 for details). For the preparation of Ni(nor), 4-($S$-acetylthio)benzaldehyde was prepared in 80% yield according to the literature method by Lindsey and colleagues[44]. The formyl group of $p$-(methylthio)benzaldehyde was protected by treatment with neopentyl glycol[45]. The methylthio group was converted into the acetylthio group upon the reaction with the methanethiolate anion followed by the introduction of acetyl chloride. Deprotection of the acetal group with trifluoroacetic acid at $-10$ °C afforded 4-($S$-acetylthio)benzaldehyde that was transformed to $p$-($S$-acetylthio)phenyl-2,2′-dipyrromethane in ∼30% yield by treatment with pyrrole. Bromination with NBS at $-75$ °C followed by DDQ (2,3-dichloro-5,6-dicyanobenzoquinone) oxidation gave 1,9-dibromo-5-($S$-acetylthio phenyl)dipyrrin in 67%. The work-up was conducted using neutral or weak acidic conditions while avoiding alumina filtration. The conventional protocol involving the use of aqueous NaOH solutions and alumina filtration was found to produce a large number of side-products including the deprotected merchaptophenyl-substituted dipyrrin. The dipyrrin was metalated in the reaction with Ni(OAc)$_2 \cdot$ 4H$_2$O and purified by recrystallization from CH$_2$Cl$_2$/hexane to get greenish black crystals in 79% yield. Finally, the

dibromodipyrrin Ni(II) complex was converted to Ni(nor) by Ni(0)-mediated homo-coupling in 10% yield. For the synthesis of Ni(porph), the 5,15-di(S-acetyl thiphenyl)-10,20-di(3,5-dimethoxyphenyl)porphyrin was prepared by the trifluoroacetic acid catalysed condensation of S-acetylthiobenzaldedhyde and 3,5-dimethoxyphenyl-2,2′-dipyrromethane. The free-base porphyrin was then metalated with $Ni(acac)_2$ to obtain Ni(porph).

**Conductance measurements.** Molecular conductance was measured using the STM-BJ technique on a Nanoscope V (Bruker, Santa Barbara, CA, USA) running a custom-made dataflow program. Scanners featuring $10\,nA\,V^{-1}$ preamplifiers were used throughout. The tunnelling current ($I$) is measured between a gold tip and a gold-on-mica substrate while they are repeatedly brought into and out of contact ($23.3\,nm\,s^{-1}$) at a constant voltage bias of 100 mV. In the absence of molecules bridging the tip–substrate gap an exponential current decay is recorded. On the contrary, if a molecule gets suspended between tip and substrate a characteristic region of constant current (plateau) is observed. Current values are transformed into conductance ($G = I/V$) and referenced to the conductance quantum ($G_0 = 2e^2/h = 77{,}481\,nS$). The Au(111) substrates were prepared by means of thermal deposition at 625–675 K under vacuum, and flame annealed before use. Tips were prepared by mechanical cutting of commercially available Au wire ($>99\%$, ø–0.3 mm). Molecular adsorption was achieved by casting a small aliquot ($\sim 20\,\mu l$) of 0.1 mM $CH_2Cl_2$ solutions of the target molecules on freshly annealed substrates. Conductance measurements were performed before and after molecular deposition. Measurements were repeated until a statistically significant data set was obtained.

**Current–voltage measurements.** The aforementioned BJ protocol was modified to record the $I$–$V$ response of individual molecules. Upon formation of a single-molecule junction, the tip was briefly held in position and the voltage scanned ($\pm 1.0\,V$ in 2.5 ms). A detailed description of the experimental protocol as well as the analysis and fitting of the $I$–$V$ profiles to a single-level tunnelling model is included in the Supplementary Information.

**Electrochemical conductance modulation.** Electrochemical control on the single-molecule conductance of Ni(nor) was exerted using a 4-electrode setup controlled by a Bruker Universal Bipotentiostat. The BJ studies were performed at different electrochemical potentials, with respect to a Ag/AgCl reference, while holding a constant tip–substrate voltage bias of 100 mV. A purpose-built three-dimensional printable cell was used to accommodate the four-electrode setup (See Supplementary Fig. 20). Gold tips were electrochemically etched in 3 M NaCl and coated in polydimethylsiloxane following a dipping/baking procedure[46]. Substrates and molecular deposition were performed following the aforementioned procedure. Experiments were performed in a pH 7 aqueous electrolyte (0.1 M, $NaH_2PO_4/Na_2HPO_4$).

**Transmission calculations.** We carried out first-principles simulations based on DFT nonequilibrium Green's functions to calculate the electronic and transport properties[47,48]. Exchange correlation was described using the generalized gradient approximation in the Perdew–Burke–Ernzerhof implementation. Au atoms were described using a single-zeta polarized basis while a double-zeta polarized basis was used for molecular atoms. Both molecules bind to tip structures via the S lone pair. In the case of the aromatic compound, we removed the side dimethoxy phenyl groups not connected to the electrodes and replaced them with H atoms to minimize the computational cost. The electronic coupling between these side groups and the central aromatic core is weak due to the large dihedral angle ($\sim 110°$) between them. In a large unit cell (42 Au atoms per layer) we verified that the projected density of states on the central aromatic core plus those binding conjugated linkers was essentially unchanged upon removal of the dimethoxy phenyl groups. We used a unit cell with 16 Au atoms per layer and optimized the vertical separation between the electrodes, relaxing the position of the molecular and tip atoms until all forces were $<0.02\,eV\,Å^{-1}$. We then carried out transmission calculations at optimized geometries. Monkhorst-Pack grids ($5 \times 5 \times 1$ and $15 \times 15 \times 1$) were used for the calculations of the electronic structure and transmission spectra, respectively. Transmission eigenchannels were computed at the centre of the Brillouin zone[49].

**Data availability.** The data that support the findings of this study are available from the corresponding authors on request.

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

## Acknowledgements

S.M.-G. gratefully acknowledges an International Research Fellowship from the Japan Society for the Promotion of Science. This work was financially supported by Grants-in-Aid for Scientific Research in Innovative Areas (Nos. 26102013, 2511008, 26102002) and Grants-in-Aid for Scientific Research (A) (No. 21340074), (B) (No. 26288020), (C) (No. 25410124) from the Ministry of Education, Culture, Sports, Science and Technology (MEXT) of Japan. N.P.A. and H.V. gratefully acknowledge financial support from the Czech Science Foundation (GAČR) under project 15-19672S and from the Purkyně Fellowship program. We thank CESNET LM2015042 and CERIT Scientific Cloud LM2015085, under the programme 'Projects of Large Research, Development, and Innovations Infrastructures' for computational resources.

## Author contributions

S.F., T.N., M.K, and H.S. conceived and designed the experiments. S.M.-G. performed the single-molecule experiments and the data analysis. T.M. performed cyclic voltammetry measurements. J.-Y.S. and H.S. synthesized and characterized the organometallic complexes. N.P.A and H.V. performed the computational studies. H.V. designed the calculations. All authors contributed to the discussion of the results. S.M.-G., S.F. and M.K. wrote the paper with contributions from all authors.

## Additional information

**Competing interests:** The authors declare no competing financial interests.

