## [Peer Review File · Nature Communications]

Reviewers' comments:

Reviewer #1 (Remarks to the Author):

This is an excellent paper, with perfect citations and convincing work. The problem is very important and this is the first successful demonstration that the increased conductivity predicted by Breslow for antiaromatic compounds has been found.

Reviewer #2 (Remarks to the Author):

Fujii et al. have synthesized a norcorrole-based complex with antiaromaticity and then investigated its electron transport properties against with its aromatic counterpart porphyrin-based complex. They revealed from experiment that the antiaromatic complex has a conductance with one order of magnitude larger than that of the aromatic one. The experimental measurements combined with the first principles calculations attributed this to the closer dominated energy level of the complex with Fermi energy of electrodes. Furthermore, they demonstrated that conductance of the antiaromatic complex can be further modulated by a factor of 5 by means of electrochemical gating in an aqueous electrolyte. This work is interesting and they have successfully provided another stable platform to investigate the relationship between antiaromaticity and conductance. This work is probably publishable, but several questions should be considered before it is accepted.

1. In the introduction, the highly related work, such as [J Am Chem Soc 2014, 136(3): 918-920] and [Chem Phys Lett 2015, 639: 131-134] should be reviewed and commented explicitly in the context. Indeed, this work is not the first one to investigate this topic. For example, Breslow et al. have already predicted this phenomenon and verified it from their experiment [Tetrahedron Letters, 2015, 56, 4833-4835], which also should be added to the context as the background of this work.

2. As is mentioned on page 19 and shown in Figure S12 of the supporting materials "we find that only one transmission channel is dominant at the Fermi level and thus current is carried by a single channel", so how the antiaromaticity open new transmission paths should be discussed in the context or in the supporting materials. At this moment, the title "Opening new transmission paths with antiaromaticity" is confusing.

Reviewer #3 (Remarks to the Author):

The authors study the electron transport properties of an antiaromatic macrocycle (Ni(nor)) and compare it with an aromatic macrocycle (Ni(porph)) using both experiment and theory. Both the experimental work and calculations seem to be carefully performed and I find no fault with the work from a technical point of view.

The issue I do have is with the story. The authors attribute the difference between the systems to antiaromaticity, but the difference they show is simply the position of the molecular levels relative to the Fermi energy. They state that the relative position can be understood as coming from the destabilization of antiaromaticity, but this is only briefly mentioned. If there is a solid basis for this idea in the literature (that the molecular levels will be necessarily closer to the Fermi energy in an antiaromatic compound), the authors should introduce this idea more carefully and pedagogically (with appropriate references). If there is no solid basis for this in the literature, then I do not believe the authors have proven that antiaromaticity is the critical feature that increases conductivity.

If we take the title as simple statement of the hypothesis, "Opening new transmission paths with antiaromaticity", I remain entirely unconvinced that the authors have proven their case. There is

certainly no evidence here that “new transmission paths” are opened with antiaromaticity.

With only two compounds studied (rather than a whole series of aromatic and antiaromatic molecules), the authors have to do more than just assert that antiaromaticity is the critical element in increasing conductivity. Without this aspect to the story, I do not believe this manuscript is of the standard for publication in Nature Communications, so I urge the authors to carefully consider the logical aspects to proving their hypothesis.

RESPONSE TO REVIEWER 1:

(Comment)

This is an excellent paper, with perfect citations and convincing work. The problem is very important and this is the first successful demonstration that the increased conductivity predicted by Breslow for antiaromatic compounds has been found.

(Response)

We thank the Reviewer for the comments. We are very happy that the Reviewer acknowledges the significance of the paper for the molecular transport community. In the introduction we have explicitly mentioned Breslow's early prediction of the higher conductance of antiaromatic compounds.

RESPONSE TO REVIEWER 2:

This work is interesting and they have successfully provided another stable platform to investigate the relationship between antiaromaticity and conductance. This work is probably publishable, but several questions should be considered before it is accepted.

(Comment 1)

In the introduction, the highly related work, such as [J Am Chem Soc 2014, 136(3): 918-920] and [Chem Phys Lett 2015, 639: 131-134] should be reviewed and commented explicitly in the context. Indeed, this work is not the first one to investigate this topic. For example, Breslow et al. have already predicted this phenomenon and verified it from their experiment [Tetrahedron Letters, 2015, 56, 4833 2013], which also should be added to the context as the background of this work.

(Response 1)

We thank the Reviewer for the careful reading and comments to our manuscript. In the resubmitted version we discuss these references explicitly and analyze our results in the context of these references.

Chen et al. (*J Am Chem Soc* 2014, 136(3): 918-920, now ref. 17) studied the conductance of three compounds with cyclic five-membered rings and found that more aromatic compounds had a lower conductance. The authors reported that conductance correlates negatively with the resonance energy (*i.e.*, with aromaticity). No antiaromatic compounds were studied in this work.

Xie et al. (*Chem Phys Lett* 2015, 639: 131-134, now ref. 18) carried out transport calculations for the three molecules of ref. 17 and observed the same trend in conductance. This trend was shown to result from the position of the HOMO-derived resonance, which was closest to the Fermi level for the nonaromatic molecule and furthest for the most aromatic compound. See also our response to Comment 1 of Reviewer 3.

We also added two other papers from the Breslow group. In the first one (ref. 13), the authors discuss related electrochemical measurements of oxidation/reduction potentials where electron donation from electron-rich to electron-poor groups was amplified if donation passed through an antiaromatic cyclobutadiene ring. The authors state how, with no source and drain, this was not “a true continuous conduction process”, although it stimulates research in realizing antiaromatic molecular circuits.

The second paper from the Breslow group that we have added was suggested by the Reviewer (*Tetrahedron Letters*, 2015, 56, 4833 2013, now ref. 19) and presents an overview of charge transport and aromaticity. In particular it reports the observation (*J Am Chem Soc* 2014, 136(3): 918-920, now ref. 17) that aromaticity diminishes conductance. We believe this paper ties very

nicely with our manuscript. At the end of the paper, just before the Conclusions section, ref. 19 states that “the argument that antiaromaticity should increase [conductance] seems reasonable. However, the proposed effect remains to be demonstrated”. We believe our paper represents the first demonstration of this phenomenon.

(Comment 2)

As is mentioned on page 19 and shown in Figure S12 of the supporting materials we find that only one transmission channel is dominant at the Fermi level and thus current is carried by a single channel, so how the antiaromaticity open new transmission paths should be discussed in the context or in the supporting materials. At this moment, the title Opening new transmission paths with antiaromaticity is confusing.

(Response 2)

We agree with the referee that the title was misleading. The title of our manuscript was changed to "Highly-conducting molecular circuits based on antiaromaticity". We believe that this study is the first demonstration that the increased conductivity predicted by Breslow for antiaromatic compounds and provides an approach to “Highly-conducting molecular circuits based on antiaromaticity”.

RESPONSE TO REVIEWER 3:

(Comment 1)

The issue I do have is with the story. The authors attribute the difference between the systems to antiaromaticity, but the difference they show is simply the position of the molecular levels relative to the Fermi energy. They state that the relative position can be understood as coming from the destabilization of antiaromaticity, but this is only briefly mentioned. If there is a solid basis for this idea in the literature (that the molecular levels will be necessarily closer to the Fermi energy in an antiaromatic compound), the authors should introduce this idea more carefully and pedagogically (with appropriate references). If there is no solid basis for this in the literature, then I do not believe the authors have proven that antiaromaticity is the critical feature that increases conductivity.

With only two compounds studied (rather than a whole series of aromatic and antiaromatic molecules), the authors have to do more than just assert that antiaromaticity is the critical element in increasing conductivity. Without this aspect to the story, I do not believe this manuscript is of the standard for publication in Nature Communications, so I urge the authors to carefully consider the logical aspects to proving their hypothesis.

(Response 1)

We thank the Reviewer for the insightful comments. From the comments of the Reviewer we realize we should explain the relationship between (anti)aromaticity and level alignment more clearly. In the introduction of the revised version we carefully explain how antiaromatic compounds have smaller HOMO-LUMO gaps than aromatic molecules, consistent with their higher reactivity and instability (refs. 6 and 7). In our case, the reduction of the gap when going from the aromatic to the antiaromatic species is seen in the MPSH orbitals in Fig. S13. The gap of the antiaromatic compound is approximately one quarter of the value for the aromatic molecule. Since the widths of the LUMO peaks, given by the electronic coupling T , are similar for Ni(nor) and for Ni(porph) on the scale of the gap values, a reduction of the gap naturally leads to a resonance position closer to the Fermi level for the species with the smaller gap.

Chen *et al.* (now ref. 17) found that more aromatic compounds had a lower conductance. The authors used the resonance energy of each species as measure of its degree of aromaticity. They examined cyclopentadiene (nonaromatic), furan (aromatic) and thiophene (the most aromatic) units and observed that cyclopentadiene had the highest conductance while thiophene had the lowest. The paper reported that the HOMO position (which governs conductance in these molecules) was further from the Fermi level for aromatic compounds than for the nonaromatic one. The paper also established that conductance correlates negatively with the resonance energy (*i.e.*, with aromaticity).

Xie *et al.* (now ref. 18) performed DFT-NEGF transport calculations for the three compounds of Chen *et al.* and reproduced the trend in conductance measured in experiment. The HOMO-derived

transmission peak was closest to the Femi level for (nonaromatic) cyclopentadiene and furthest for (most aromatic) thiophene. This trend was consistent with the calculated positions of molecular eigenstates. The value of the gap defined by the HOMO- and LUMO-derived resonances also decreased with increasing aromatic character.

We have rewritten the introduction to make these elements clear in the context of these references. Our result of higher conductance arising from a closer LUMO position in the antiaromatic species is consistent with these previous findings, observed here for the first time in a purely antiaromatic compound. In addition, simple theoretical calculation and discussion on the reduced gap of antiaromatic molecules are added to Supplementary information (SI 11).

(Comment 2)

If we take the title as simple statement of the hypothesis, Opening new transmission paths with antiaromaticity, I remain entirely unconvinced that the authors have proven their case. There is certainly no evidence here that new transmission paths are opened with antiaromaticity.

(Response 2)

We agree with the referee that the title was misleading. The title of our manuscript was changed to "Highly-conducting molecular circuits based on antiaromaticity". We believe that this study is the first demonstration that the increased conductivity predicted by Breslow for antiaromatic compounds and provides an approach to "Highly-conducting molecular circuits based on antiaromaticity".

REVIEWERS' COMMENTS:

Reviewer #2 (Remarks to the Author):

The authors have revised the manuscript in a satisfactory manner and therefore this paper is recommended for publication.

Reviewer #3 (Remarks to the Author):

The authors have responded convincingly to all the points I raised and in my opinion the manuscript is now ready for publication.

RESPONSE TO REVIEWER 2:

(Comment)

The authors have revised the manuscript in a satisfactory manner and therefore this paper is recommended for publication.

(Response)

We thank the Reviewer for the comment. We are very happy that the Reviewer acknowledges the significance of the paper.

RESPONSE TO REVIEWER 3:

(Comment)

The authors have responded convincingly to all the points I raised and in my opinion the manuscript is now ready for publication.

(Response)

We thank the Reviewer for the comments. We are very happy that the Reviewer acknowledges the significance of the paper.